# Distribution of Nitrate Content in Groundwater and Evaluation of Potential Health Risks: A Case Study of Rural Areas in Northern China

**DOI:** 10.3390/ijerph17249390

**Published:** 2020-12-15

**Authors:** Wenwen Feng, Chao Wang, Xiaohui Lei, Hao Wang, Xueliang Zhang

**Affiliations:** 1School of Water and Environment, Chang’an University, Xi’an 710054, China; fww@chd.edu.cn; 2Key Laboratory of Subsurface Hydrology and Ecological Effect in Arid Region of Ministry of Education, Chang’an University, Xi’an 710054, China; 3State Key Laboratory of Simulation and Regulation of Water Cycle in River Basin, China Institute of Water Resources and Hydropower Research, Beijing 100038, China; lxh@iwhr.com (X.L.); Wanghao@iwhr.com (H.W.); 4Department of Water Resources, China Institute of Water Resources and Hydropower Research (IWHR), Beijing 100038, China; 5Inner Mongolia Land and Resources Exploration and Development Institute, Hohhot 010020, China; nmgzxl2020@163.com

**Keywords:** mine water, groundwater quality, nitrate pollution, human health risk, Inner Mongolia

## Abstract

Nitrate pollution is considered to be one of the most common environmental problems in groundwater, especially in areas affected by human mining, such as the arid region of northern China. However, the human health risk assessment of nitrate pollution in this area has not yet been carried out. In this study, groundwater samples were taken in the Selian mining area in Inner Mongolia to conduct a full analysis of water quality. On this basis, the groundwater quality, the distribution range of nitrate pollution, and human health risks were evaluated. The results show that the groundwater in the Selian mining area is neutral to alkaline, with high salinity and hardness. The concentration of nitrate ions in groundwater generally exceeds the standard, and the maximum exceeds 5.48 times the value specified in the Chinese national standard, indicating that groundwater nitrate pollution needs to be controlled urgently. Groundwater is polluted by large amounts of nitrogen fertilizer used by humans in agricultural activities. At the same time, mining activities have accelerated the severity and spread of pollution. Groundwater is not recommended for direct human life and irrigation use in the study area unless purification measures are taken. Nitrate pollution is more harmful to children through groundwater, about 1.54 times that of adults. Excess nitrate is transported into the body through drinking groundwater, so proper drinking water control will reduce the health risks of nitrate, such as centralized water supply. This study will provide a scientific basis for the rational use of groundwater and nitrate pollution control in the area.

## 1. Introduction

With the rapid development of the social economy, the natural environment is constantly being polluted by humans [1,2,3]. Groundwater is an important resource for human survival, and it is also an important consumable for industrial development such as coal mining in the arid and semi-arid regions of Northwest China [4,5,6]. Environmental pollution of groundwater and deterioration of water quality continue to occur because of long-term coal mining [7,8,9]. Groundwater is an important part of the hydrological cycle and plays a vital role in maintaining biodiversity and geochemistry [10,11,12]. Unfortunately, humans have used chemical fertilizers intensively for a long time, making groundwater in many areas face severe nitrate pollution [13,14,15]. Coal has always been the main supplier of energy in China. Seeking harmony between coal mining and other human activities and groundwater environmental protection is of great significance for ensuring the sustainable development of the local economy and national energy security [16,17].

The coal seams in northern China mainly exist in the Jurassic strata, which contains several aquifers [3,18,19]. The mining of these coal seams will affect the quantity and quality of groundwater in the overlying aquifer, increase the circulation of groundwater, and cause water inrush and water pollution [20]. The problem was reported in water inrush disasters and mine water pollution due to coal mining. This threatens the safety of local residents’ mining and water supply [20,21]. For example, He [21] studied groundwater pollution caused by coal mining in Shanxi, China. Coal mining will bring a large amount of coal powder and rock debris into the soil and aeration zone, polluting pore water. The next step is to pollute groundwater through the driving force of agricultural irrigation and channel infiltration.

Nitrate pollution in groundwater has been a major environmental problem worldwide [22,23]. It is generally believed that groundwater generally has a higher quality. However, there will be a certain connection channel between the surface and groundwater. Because of human over-mining activities, groundwater is more susceptible to the influence of surface land use and infiltration and enters from the surface [24,25]. What is more serious is that humans have to use contaminated groundwater because of the shortage of water resources. This will cause a vicious circle to put your health at risk [26,27]. According to research, high concentrations of nitrate not only pose a serious threat to aquatic ecosystems but also may affect human health through drinking water [28]. Nitrate itself does not directly harm the human body, but when converted into nitrite in the human body, it can induce certain diseases, such as infant methemoglobinemia and cancer. The concentration of nitrate in drinking water is limited to less than 20 mg/L, which is included in the National Drinking Water Standards for Residents in China [29]. The United States also has a similar limit of 50 mg/L [30]. Therefore, the nitrate concentration must be monitored for early warning of nitrate pollution in the groundwater. In this way, accurate health risk assessment can be carried out and it provides a reference for groundwater management and pollution control.

Selian mining area is located in the center of Ordos Energy and Heavy Chemical Industry Base, a national-level energy and heavy chemical industry base. The region is rich in mineral resources and the economy is relatively developed [31]. In recent years, the amount of groundwater extraction has gradually increased due to the influence of human activities. The climate in the region continues to tend to be arid, and the groundwater environmental problems are very prominent [31,32]. Many scholars point out that the existing extensive mineral resource development methods have caused serious damage to the ecological environment in some areas. Then, the methods of tackling environmental problems are simple and rough, and they do not pay attention to real-time monitoring and early warning. They often take measures passively after major accidents, causing great personnel and economic losses [31,32]. In addition, due to geological factors, the chemical characteristics of groundwater are complex and changeable, which is greatly affected by external human activities. Although the mining of coal mines in this area has attracted the attention of the local water resources management department, a comprehensive evaluation of groundwater quality and human health risks has not yet been conducted [32,33]. Understanding the quality of groundwater and the degree of nitrate pollution in this area is essential for future groundwater management and pollution control. 

The human health risk assessment model proposed by the US Environmental Protection Agency is one of the most useful methods to quantify the potential risks of groundwater pollutants. It has been widely used to assess the potential health risks caused by water pollutants and provide a scientific basis for the local water supply department. Therefore, this paper selects 20 groundwater sample points around the mining area to conduct a full analysis of water quality. The main purpose of this research is to: (1) evaluate the quality of groundwater in coal mining areas; (2) study the situation and sources of nitrate pollution; (3) determine the impact of nitrate pollution on human health risks and analyze its characteristics.

## 2. Study Area

### 2.1. Location and Climate

Selian mining area is located in Dongsheng District, Ordos City, Inner Mongolia. It is 13 km east of Dongsheng and extends between longitude 109°43′09″~109°49′50″ and latitude 39°52′27″~39°59′50″ (Figure 1). National Highway 109 passes by, and the transportation network extends in all directions. The elevation in the south is higher than in the north and the village is located in the north. The highest point is located in the south of the study area, with an elevation of 1504.0 m. The lowest point is located in the northeast of the study area, with an elevation of 1391.5 m and a maximum elevation difference of 112.5 m. Due to human mining activities, the original landforms in the study area are seriously damaged, and the valleys are vertical and horizontal, which is a typical erosive hilly landform. 

The climate is moderately temperate, semi-arid, and semi-desert monsoon climate. Winter is cold and arid, with little rainfall, and summer is hot and evaporates. The annual average temperature of the study area is 6.4 °C, the highest temperature is 36.7 °C, and the lowest temperature is −31.5 °C. The four seasons are hot and cold, the temperature difference between day and night is huge, drought and rain are low, and evaporation is large. The distribution of precipitation throughout the year is extremely uneven, with rainfall mostly concentrated in July, August, and September, accounting for 66% of the annual precipitation. The average rainfall for many years is 276.2 mm, the rainfall in the wet year can reach 819.0 mm, and the rainfall in the dry year can reach 108.6 mm. The freezing period is long throughout the year, generally from the beginning of October to the end of April of the following year. The average evaporation for many years is 2494.0 mm, which is much higher than rainfall. There are no perennial surface water runoffs and lakes around the study area, but mainly seasonal rivers. Atmospheric precipitation is the main source of groundwater supply. In addition, there are several valleys formed by rainfall erosion in the northern part of the study area. The seasonal flow of these valleys is mainly recharged by precipitation and local spring water. The shortage of surface water available for domestic use makes groundwater in the area important.

### 2.2. Hydrogeology

The formation and distribution of groundwater in this area are controlled by geography and geological environment. According to the lithology, thickness, burial conditions, and spatial distribution of the stratum, four different aquifers are classified within the depth of the study area. The unconfined aquifer formed in the Quaternary loess has no stable aquifer between the underlying Cretaceous aquifer rock series, so it forms a unified phreatic aquifer with the Cretaceous aquifer rock series. A confined fractured aquifer formed in the Jurassic Anding Formation and Zhiluo Formation geological unit.

The phreatic aquifer is mainly distributed in the southern part of the study area. The lithology is variegated sandy gravel, silty fine sand layer, with loose consolidation, well-developed pores, unit water inflow q = 1~5 L/s·m, and strong water richness. The burial depth of diving is generally less than 10m, and the burial depth in the deep section of the valley is larger, up to 30m. In alluvial beaches and around lakes and swamps, diving depth is the shallowest, less than 1m. The main source of replenishment for diving is atmospheric precipitation, followed by lateral runoff replenishment, sand dune condensate water, and deep confined water replenishment in suitable areas. Due to the small amount of precipitation and a large amount of evaporation, the recharge of diving is also small. Diving runoff is mainly controlled by geomorphic conditions. The burial of diving is controlled by factors such as terrain undulations and valley cutting depth.

Confined water aquifers are widely distributed, and the water richness is controlled by the lithology and thickness of the aquifer. The unit water inflow is generally 0.25–1.00 L/s·m, and locally it can reach 5.5 L/s·m, and the water richness is weak. The main source of confined water recharge is lateral runoff recharge. The second is to receive the infiltration and replenishment of atmospheric precipitation at the surface outcrops. In the higher terrain, it also accepts the leakage flow for diving. The general flow direction of confined water is runoff from northwest to southeast, and the hydraulic gradient is generally 0.0011 to 0.0021. Due to the cross-flow replenishment of confined water by highland diving, the confined water level in this area is raised, forming a local runoff of confined water from high places to low-lying places. Confined water is still mainly discharged by lateral runoff. The second is to carry out the drainage of the top diving and the drainage of artificially pumped wells.

## 3. Materials and Methods

### 3.1. Sample Collection and Analysis

The sampling time was 13–15 July 2019, during sunny days. A total of 20 groundwater samples were collected in the study area at this time. A standard 125 mL polyethylene bottle was used as the sampling bottle, and the sampling bottle was cleaned with deionized water before going out for sampling. When sampling in the field, it was rinsed with sampling water three times, put into a sampling bottle, and sealed [34]. GPS was used to locate the sampling point during sampling, and the well depth and groundwater depth were recorded on site. While sampling, pH, conductivity (EC), and water temperature (T) of the water sample were determined. The sample was sent to the water quality testing laboratory for testing and analysis as soon as possible. Among them, Na^+^ and K^+^ were tested by flame atomic absorption spectrophotometry, Ca^2+^ and Mg^2+^ were tested by EDTA titration, SO_4_^2−^ and Cl^−^ were determined by ion chromatography, and HCO_3_^−^ was determined by acid–base titration.

After analysis, the charge balance error percentage (%CBE) was calculated to check the accuracy of each sample test [1,35]. The percentage of charge balance error (%CBE) is expressed as follows:(1)$$\%\mathrm{CBE}=\frac{\sum cation-\sum anion}{\sum cation+\sum anion}\times 100\%$$

The units of anion and cation in the above formula are milliequivalents per liter.

It is generally believed that if the charge balance error percentage (%CBE) is less than 5%, the test is considered qualified and correct. The %CBE of sampling points in this research was less than 5%.

### 3.2. Improved Groundwater Quality Index

The Entropy Weighted Water Quality Index (EWQI) is an important water quality testing tool, which has been evaluated from the perspective of its practicality and management [10,36]. It obtains the water quality index by assigning certain weights to different water quality indicators and multiplying by the corresponding actual values. To reflect the impact of various parameters on the overall quality of drinking water. Its main purpose is to convert a large amount of water quality data into understandable and informative data. It provides water quality data to the public, decision-makers, and management authorities in a very simple way, which is very effective. These indicators are widely used to assess the water quality of countries around the world [37,38].

Using EWQI to characterize groundwater quality, the steps to calculate EWQI are as follows. Assuming there are m groundwater samples, and each sample has n water chemical parameters, an initial rating matrix is established:(2)$$X=\left[\begin{array}{llll} x_{11} & x_{12} & \cdots & x_{1n}\\ x_{21} & x_{22} & \cdots & x_{2n}\\ \vdots & \vdots & \ddots & \vdots \\ x_{m1} & x_{m2} & \cdots & x_{mn} \end{array}\right]$$
where $x_{ij}$ is the sample concentration, i=1,2,⋯,m,j=1,2,⋯,n.

Standardize the initial data according to Formula (3).
(3)$$\left\{ \begin{array}{l} y_{ij}=\frac{x_{ij}-(x_{ij}{)}_{\min}}{{\left(x_{ij}\right)}_{\max}-(x_{ij}{)}_{\min}}\\ y_{ij}=\frac{{\left(x_{ij}\right)}_{\max}-x_{ij}}{{\left(x_{ij}\right)}_{\max}-(x_{ij}{)}_{\min}} \end{array}\right.$$

After standardization, the rating matrix can be obtained:(4)$$Y=\left[\begin{array}{llll} y_{11} & y_{12} & \cdots & y_{1n}\\ y_{21} & y_{22} & \cdots & y_{2n}\\ \vdots & \vdots & \ddots & \vdots \\ y_{m1} & y_{m2} & \cdots & y_{mn} \end{array}\right]$$

The information entropy of the indicator can be obtained by using the following two formulas:(5)$$P_{ij}=\frac{y_{ij}}{\sum_{i=1}^{m}y_{ij}}$$
(6)$$e_{j}=-\frac{1}{lnm}\sum_{i=1}^{m}P_{ij}lnP_{ij}$$

Finally, the weight of each indicator is obtained,
(7)$$\omega_{j}=\frac{1-e_{j}}{\sum_{j=1}^{n}(1-e_{j})}$$

Sodium adsorption ratio (SAR) is an important indicator to measure the harm of sodium in irrigation water. It is the concentration of sodium relative to calcium and magnesium. Calculated according to the following standard formula [7,39]:(8)$$SAR=\frac{Na^{+}}{\sqrt{\frac{Ca^{2+}+Mg^{2+}}{2}}}$$

### 3.3. Human health risk assessment

According to the characteristics of pollutants, the water environment health risk assessment model can be divided into a genotoxic substance evaluation model and a body toxic substance evaluation model. Nitrate is a body toxic substance [22,40]. It is generally believed that the risk assessment of somatic toxic substances is based on the reference dose. When the exposure dose of the target substance exceeds the reference dose, it may produce toxic effects. The evaluation model is [41,42]:(9)$$HI=\frac{I_{CD}}{D_{Rf}}$$
where HI is a non-carcinogenic risk index (dimensionless), $I_{CD}$ is the average daily exposure dose (mg·kg^−1^·d^−1^), and $D_{Rf}$ is the reference dose of a specific non-carcinogen in groundwater (mg·kg^−1^·d^−1^).

The non-carcinogenic risk threshold recommended by the US Environmental Protection Agency [30] is 1. When *HI* < 1, the human non-carcinogenic health risk caused by the pollutant is within an acceptable range, and the health of the contact is unlikely to suffer obvious adverse effects. When *HI* > 1, it indicates that the human non-carcinogenic health risk caused by the pollutant is unacceptable, and as the HI increases, the non-carcinogenic health risk tends to increase [40,43].

Nitrate, the target pollutant in drinking water sources, enters the human body mainly through drinking water intake and skin contact. The calculation formula for the average daily exposure dose is [44,45]:(10)$$I_{CD}=I_{CDI}+I_{CDD}$$
(11)$$I_{CDI}=\frac{C\times IR\times ABS\times EF\times ED}{BW\times AT}$$
(12)$$I_{CDD}=\frac{C\times SA\times K_{p}\times EV\times ET\times ED\times CF}{BW\times AT}$$
where $I_{CDI}$ is the average daily exposure dose through drinking water (mg·kg^−1^·d^−1^); $I_{CDD}$ is the average daily exposure dose of skin contact route (mg·kg^−1^·d^−1^); C is the measured concentration of nitrate in groundwater (mg·L^−1^); IR is drinking water rate (L·d^−1^); ABS is the gastrointestinal absorption coefficient and is related to pollutants; EF is the exposure frequency, which is the time of exposure in a year; ED is the duration of exposure, indicating the number of years that the body has ingested the substance throughout its life; BW is the average weight of residents (kg); AT is the life expectancy, which is the average time the exposure occurs; SA is the skin contact surface (cm^2^); $K_{p}$ is the skin permeability coefficient of pollutants (cm·h^−1^); EV is the bath frequency, how many days do you take a bath; ET is bath time (h·d^−1^); CF is the volume conversion factor (Table 1).

## 4. Results and Discussion 

### 4.1. Groundwater Geochemistry

The characteristics of the physicochemical parameters of ions in groundwater are shown in Table 2. The groundwater in the phreatic aquifer is alkaline, and the change trend is not big. TDS (Total Dissolved Solids) is between 365.60 and 922.91 mg/L, all of which are less than 1000 mg/L, indicating that the groundwater quality in this area is good and suitable for human drinking (Figure 2a). The order of the main cation concentration in the study area is as follows: Ca^2+^ > Na^+^ > Mg^2+^ > K^+^. The total hardness (TH) is between 205.17 and 570.48, calculated as CaCO_3_. Groundwater dominated by Ca^2+^ and Mg^2+^ usually leads to higher water hardness, and boiling water contains more impurities after precipitation. At the same time, it shows that Ca^2+^ and Mg^2+^ in the groundwater in this area may mainly come from the dissolution of minerals such as carbonate and gypsum. The order of the main anion ion content is: HCO_3_^−^ > SO_4_^2^^−^ > Cl^−^ > NO_3_^−^. From the spatial distribution map of TDS concentration in Fig. 2a, the TDS in the north of the study area is higher than that in the south, and the north is where humans live. It shows that the abnormal increase in TDS is affected by human activities. Overall, the groundwater quality in this area is good, and the dissolved solid content in some areas is abnormal. It indicates that high salinity in groundwater may also be affected by groundwater evaporation and soil salt dissolution.

### 4.2. Nitrate Content in Groundwater

The concentration of nitrate in the study area is between 0.80–109.57 mg/L, with an average of 23.57 mg/L. According to China’s national drinking water sanitation standard limit of 20 mg/L (with nitrate nitrogen as the evaluation index) [29], 25% of the nitrate in the groundwater in the study area exceeded the standard, and the maximum exceeded multiple was 5.48 times. Adopting the WHO [47] drinking water standard of 50 mg/L, the nitrate content of groundwater in this area exceeded the standard by 15%, and the maximum multiple of exceeding the standard reached 2.19 times. It shows that nitrate pollution in groundwater needs to be controlled urgently. Figure 2b shows that the nitrate pollution in the northern part of the study is more serious, which is mainly because the northern part is a plain area with a large number of villages and farmland. Due to economic development and population increase, in order to increase food production, a large amount of nitrogen fertilizer will be invested in the process of agricultural activities. Nitrogen fertilizer flows into the ground with irrigation water and rainwater and is eventually converted into nitrate. Nitrate pollutants can easily infiltrate into the groundwater through the subsurface, leading to aggravated groundwater nitrate pollution. Compared with the northern regions, there are no crops or large human settlements in the south. It is understood that the local soil is poor and a lot of chemical fertilizers are used in daily planting. Due to the low level of economic development, agricultural planting can only use a large number of low-priced nitrogen fertilizers instead of high-efficiency compound fertilizers. As a result, there is a vicious circle of reduced food production—increased chemical fertilizers as a last resort—increased groundwater irrigation pollution—and soil fertility degradation [13,46].

### 4.3. Groundwater Quality Assessment

The calculation result of the entropy weighted water quality index (EWQI) of the sampling point is shown in Figure 2c. The result shows that the water sample index is between 72.90 and 137.57, and the water quality ranges from poor to good. Among them, the sampling points with good water quality account for 25%, which are SQ04, SQ05, SQ09, SQ15, and SQ20, which are suitable for human drinking. The water quality in 75% of the sampling sites is poor and it is not recommended for human consumption. This indicates that there is serious groundwater pollution in the confined aquifer. Combining the analysis results, it can be concluded that the main reason for the change in water quality at the sampling point is that agriculture or mining activities lead to the increase of nitrate concentration in the area, so the water quality index is low. In addition, the oxidation and leaching process of pyrite and coal clean waste will also affect the chemical properties of the diving groundwater [7,10].

The SAR range of all groundwater samples is between 1.46 and 10.55, with an average value of 6.25, which indicates that the groundwater in this area is suitable for irrigation in terms of alkalinity (Figure 3). Figure 3 also shows that the salinity of groundwater sample points in this area are all located in the C3 area, indicating that most of the sampling points have high salinity and are more suitable for irrigation. If you want to use groundwater in the area for irrigation, it is recommended to mix irrigation with surface water [24].

### 4.4. Health Risk Assessment

According to the definition of risk index by the United States Environmental Protection Agency [30], the acceptable level of risk for non-carcinogenic chronic toxic effects is 1. Table 3 shows the evaluation results of the health risk model. The risk index of children with drinking water intake of nitrate ranges from 0.0129 to 1.7609, with an average value of 0.3789, which is approximately 1.54 times that of adults. Among them, the risk index of more than 1 accounted for 10% and 5%, respectively, indicating that the nitrate content in the groundwater in the study area is carcinogenic and seriously threatens human health. In addition, the proportion of children and adults in the study area with a risk index between 0.5 and 1 is 15% and 10%, respectively. Although the health risk index of these samples does not exceed 1, it is very likely that an agricultural activity or unreasonable exploitation of groundwater causes its value to exceed 1, which threatens human health. Therefore, the health risks of these spots also need attention. The carcinogenic risk index through skin contact is far less than 1, indicating that the health risk of nitrate through skin contact is very small and within an acceptable range.

By comparing the non-carcinogenic risks of drinking water and skin contact, the health risks of children through drinking water intake are significantly higher than that of adults, 1.54 times that of adults. This may be due to the fact that children per unit of weight are more sensitive to environmental pollution than adults [48], so the health risks of nitrate in groundwater to children should be paid attention.

From the spatial distribution of the total risk index of the two exposure pathways (Figure 4), the northern part of the study area has the highest nitrate concentration, so all populations (children and adults) in these areas are at high risk. The non-carcinogenic health risk index of nitrate intake through drinking water is greater than 1, and the risk index for children even reaches 1.76 in some places, indicating that human health has been seriously threatened. These places are located in the northern part of the mining area and are mainly human settlements and agricultural plantations, which should attract the attention of relevant departments. From the distribution area, the area with a higher risk index for children is significantly larger than that for adults, indicating that children are more sensitive recipients and face higher health risks than adults.

It can be seen that the total risk of nitrate intake in the two exposure routes is roughly equal to the health risk of drinking water. In the total risk, the contribution rate of non-carcinogenic risk caused by drinking water accounted for 99.40%, which is much greater than that of skin contact. This shows that nitrate in groundwater enters the human body mainly through drinking water. This is consistent with the research results of Sheng [49] and Cai [45]. Due to its location in the mountains, groundwater collected from wells is used as the only source of drinking water for residents in the area. Therefore, to reduce the health risks of nitrates, drinking water control should be the main focus, such as centralized water supply. The northern part of the study area has the highest nitrate concentration, so all populations in these areas are at high risk, while the health risks in other areas are relatively low. 

## 5. Conclusions

In order to understand the impact and harm of human mining and agricultural activities on groundwater quality, field investigations and groundwater sample analysis were carried out in the Selian mining area in Ordos City, Inner Mongolia, China. The use of chemical fertilizers in mining and agriculture pollutes the groundwater environment, making groundwater quality worse. The following conclusions can be drawn:(1)The groundwater in the Selian mining area is neutral to weakly alkaline, with high salinity and medium hardness, which is more suitable for human consumption. Groundwater quality in this area is affected by weathering of rock formations, coal seams, and evaporation. Most of the samples are HCO_3_-Ca type water, the order of cations is Ca^2+^ > Na^+^ > Mg^2+^ > K^+^, and the order of anions is HCO_3_^−^ > SO_4_^2^^−^ > Cl^−^ > NO_3_^−^.(2)The concentration of nitrate in groundwater in the study area is between 0.80–109.57 mg/L, with an average of 23.57 mg/L. The largest exceeds China’s national drinking water limit standard by 5.48 times and exceeds the WHO standard by 2.19 times. It shows that nitrate pollution in groundwater needs to be controlled urgently. The input of a large amount of nitrogen fertilizer in human agricultural activities is the main source of pollution, and at the same time, mining activities accelerate the severity and spread of pollution. The groundwater quality assessment based on EWQI shows that the groundwater in this area is not suitable for direct drinking by humans. Nitrate pollution is the main physical and chemical parameter leading to poor water quality. In addition, groundwater is not suitable for direct irrigation. Unless pretreatment is possible, other water sources should be studied for irrigation.(3)The results of the human health risk assessment model show that about 10% of the groundwater’s non-carcinogenic chronic toxicity effects in this area are at an unacceptable level for children. The nitrate health risks of children through drinking water intake and skin contact are significantly higher than adults, and the highest is 1.54 times. The risk index is between 0.75 and 1 and the proportion is about 15%. It is in a period of dangerous fluctuations and is susceptible to changes in the external environment and endangers human health. Therefore, these health risk points require special attention.

## Figures and Tables

**Figure 1 ijerph-17-09390-f001:**
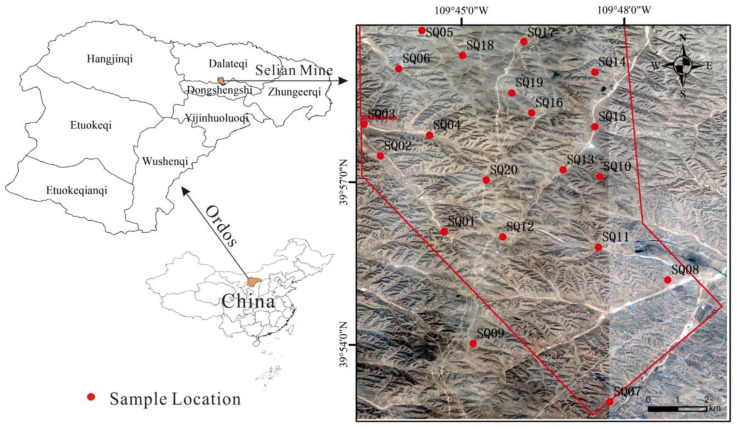
Location of the study area.

**Figure 2 ijerph-17-09390-f002:**
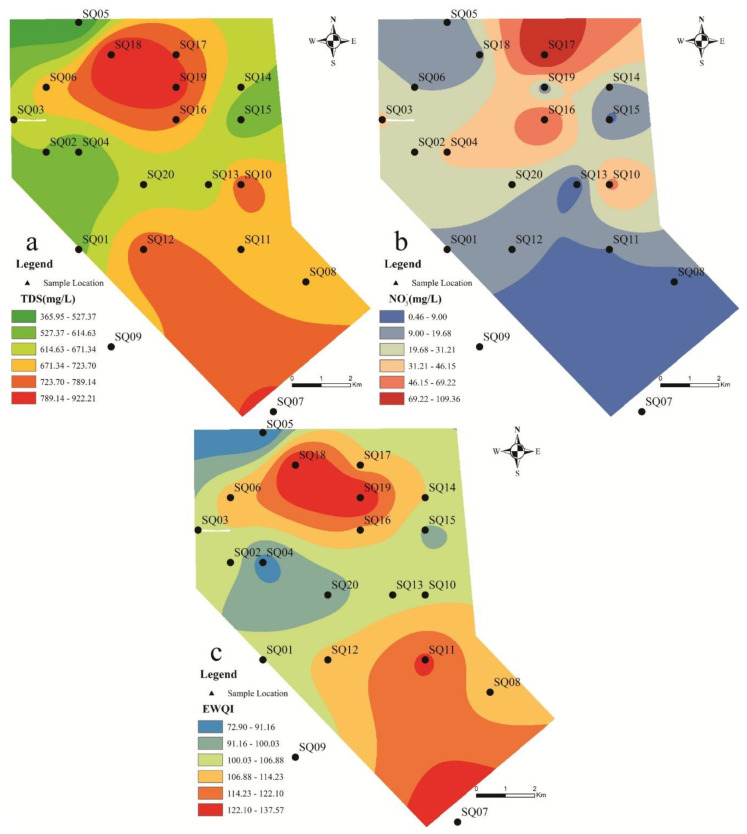
TDS (Total Dissolved Solids) and NO_3_^−^ spatial distribution map of the study area.

**Figure 3 ijerph-17-09390-f003:**
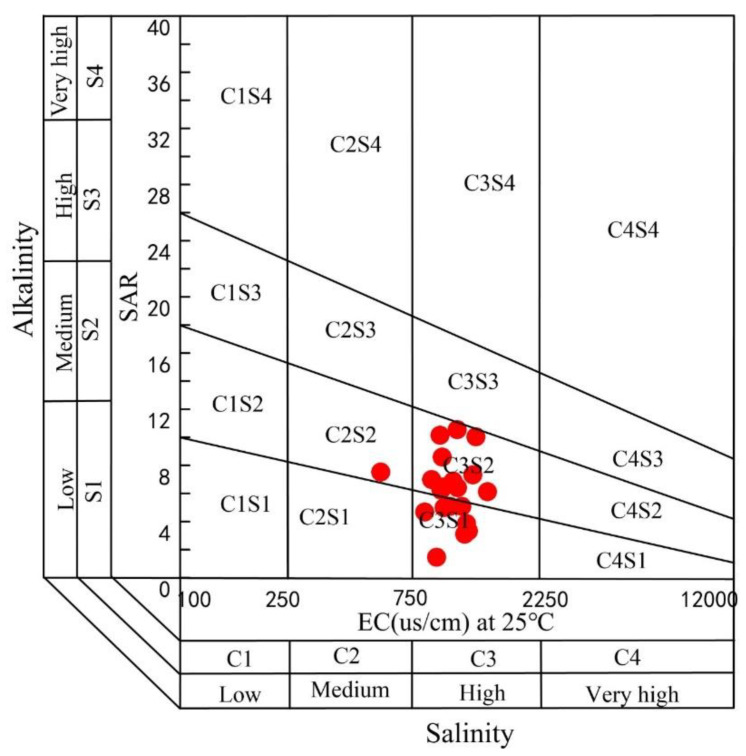
USSL (US Salinity Laboratory) diagram for evaluating the suitability of groundwater irrigation.

**Figure 4 ijerph-17-09390-f004:**
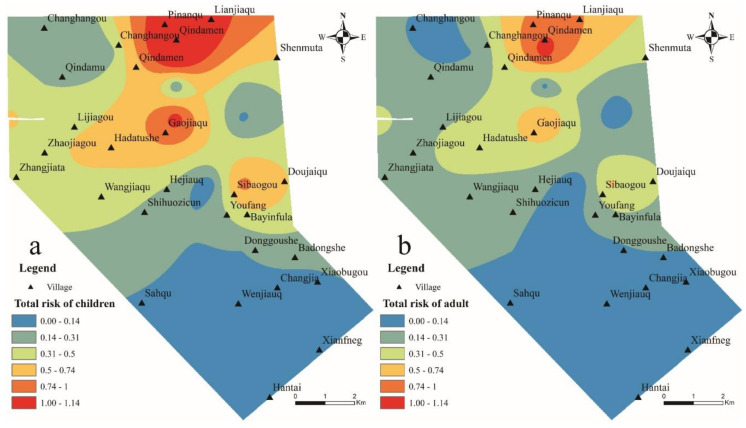
Spatial distribution of non-carcinogenic risk of nitrate.

**Table 1 ijerph-17-09390-t001:** Parameters employed for human health risk assessment [42,44,46].

Parameters	Children	Adult
$D_{Rf}$ (Nitrate reference dose)/(mg·kg^−1^·d^−1^)	1.6	1.6
C (Nitrate concentration)/(mg·L^−1^)	Measured	Measured
IR (Drinking rate)/(L·d^−1^)	1.8	2.0
BW (Average weight of residents)/kg	35	60
ET (Bath time)/(h·d^−1^)	0.167	0.167
ABS (Gastrointestinal absorption coefficient)	0.5	0.5
AT (Life expectancy)/a	365 × ED	365 × ED
EV (Bathing frequency)	1	1
$K_{p}$ (Skin permeability coefficient)/(cm·h^−1^)	0.001	0.001
CF (Volume conversion factor)/(L·cm^−2^)	1/1000	1/1000
ED (Exposure duration)/a	30	30
EF (Exposure frequency)/(d·a^−1^)	365	365
SA (Skin contact surface area)/cm^2^	1.0 × 10^4^	1.65 × 10^4^

**Table 2 ijerph-17-09390-t002:** Table of main statistical characteristics of research groundwater.

Index	Max	Min	Mean	Standard Deviation	Coefficient of Variation
pH	8.48	7.80	8.11	0.19	0.02
TH	570.48	205.17	421.85	92.17	0.22
TDS	922.91	365.60	678.11	120.02	0.18
Ca^2+^	118.24	42.08	78.06	21.28	0.27
Mg^2+^	71.70	24.31	55.11	11.74	0.21
K^+^	4.30	0.39	1.15	0.82	0.71
Na^+^	163.01	32.65	77.71	32.42	0.42
Cl^−^	177.27	35.45	81.72	38.34	0.47
HCO_3_^−^	524.67	183.02	330.36	87.87	0.27
SO_4_^2−^	288.00	57.60	186.00	49.86	0.27
NO_3_^−^	109.57	0.80	23.57	26.16	1.11

**Table 3 ijerph-17-09390-t003:** Non-carcinogenic risk of nitrate for children and adults in drinking water and Dermal contact pathway.

Non-Carcinogenic	Water Intake Risk Index		Skin Contact Risk Index		Total Risk of Two Exposure Routes	
	Range	Average	Range	Average	Range	Average
Children	0.0129–1.7609	0.3789	0.0000–0.0033	0.0007	0.0129–1.7642	0.3796
Adult	0.0083–1.1413	0.2456	0.0000–0.0031	0.0007	0.0084–1.1445	0.2462

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
