# Peer review of "Distribution of Nitrate Content in Groundwater and Evaluation of Potential Health Risks: A Case Study of Rural Areas in Northern China"

_ijerph, 2020, doi:10.3390/ijerph17249390_

Round 1

Reviewer 1 Report

Dear Authors, I propose to change the character of yours manuscript from article to report. If the obtained results of health risk based only are on one measurement in time in my opinion it can’t provide to present general conclusions. The many factors can be responsible for the higher nitrite concentration like for example industry or bad waste water management The research are interesting and good presented. I recommended to show satellite view on the study area. The legends on figures should be greater. Figure 4 should present the health risk on the map base with the indication of inhabited places line 96-97 you informed about elevation so you mean m above water level? Line 332 what are the sources of water supply in a given area How many people are connected to central water supply How many people using wells (how big is a problem) Materials and method have to be detailed” 1 How many samples were collected from each appointed measurements points? If only ones the conclusions cannot be so determined 2 What was the time of sampling for example at this same time you mean this same day. 3 At what time of year were the samples taken? what agricultural works were performed at that time 4 Explain please the weather condition before and during the sampling Conclusions It is important to explain why authors suggested that the main source of pollutant comes from nitrogen fertilizer used in agricultures? It is necessary to describe waste water management in this area to exclude the waste water influence.

Author Response

Response to Reviewer 1 Comments

Point 1: I propose to change the character of yours manuscript from article to report. If the obtained results of health risk based only are on one measurement in time in my opinion it can’t provide to present general conclusions. The many factors can be responsible for the higher nitrite concentration like for example industry or bad waste water management .

Response 1: Thank you very much for your valuable comments. After careful consideration, I decided to maintain the status quo.

Point 2: I recommended to show satellite view on the study area. The legends on figures should be greater.

Response 2: Modified, see Figure 1.

Point 3: Figure 4 should present the health risk on the map base with the indication of inhabited places line 96-97 you informed about elevation so you mean m above water level?

Response 3: Modified, see Figure 4.

Point 4: Line 332 what are the sources of water supply in a given area How many people are connected to central water supply How many people using wells (how big is a problem)

Response 4: Groundwater, without a centralized water supply, it is also groundwater extracted. Groundwater collected from wells is the only source of drinking water for residents in this area.

Point 5: How many samples were collected from each appointed measurements points? If only ones the conclusions cannot be so determined.

Response 5: Each village collected a 500ml bottle of mineral water, went back to test three times to get the average value, and did precision analysis to ensure that the experimental results are accurate and reliable.

Point 6: What was the time of sampling for example at this same time you mean this same day.

Response 6: The sampling time is July 13-15, 2019, normal working hours.

Point 7: At what time of year were the samples taken? what agricultural works were performed at that time

Response 7: The sampling time is July 13-15, 2019. A large area of wheat is planted in the northern area of the study area. The sampling time is about one month after the local wheat harvest. In the southern area of the study area, there are no crops on the ground, only scattered weeds and trees.

Point 8: Explain please the weather condition before and during the sampling

Response 8: The weather on the sampling day was sunny.

Point 9: Conclusions It is important to explain why authors suggested that the main source of pollutant comes from nitrogen fertilizer used in agricultures? It is necessary to describe waste water management in this area to exclude the waste water influence.

Response 9: Because the nitrate pollution in the northern part of the study area is obviously serious, while the southern part lacks a low nitrate concentration. Compared with the north, there are no crops and no human settlements in the south, and it is understood that the local soil is poor and uses a lot of chemical fertilizers in daily planting. Due to the low level of economic development, agricultural planting can only use low-priced nitrogen fertilizers. Not a highly efficient compound fertilizer. Therefore, it is concluded that agricultural activities are the main source of local groundwater pollution.

Reviewer 2 Report

The manuscript is a well-written contribution to the field; some points should be covered before continuing with the assessment

1) Introduction did not indicate the state of the art in the field, and should be improved with more references and indicating the gaps in the knowledge and also the main novelty of the work; in this points, please differentiate with the previously reported article from a reference

2) In the material and method section, the3.2. Improved Groundwater Quality Index is not clear, please improve and also explain more its utility

3. Some results are not clear the way they were calculates, for example

 230-233 "25% of the nitrate in the groundwater in the study area exceeded the standard",.... " the nitrate content of groundwater in this area exceeded the standard by 15% "

in row 271-272 "...indicating that the nitrate content in the groundwater in the study area is carcinogenic and seriously threatens human health" I do not understand well if the formula is non-carcinogenic compounds how the authors express that the compound is carcinogenic, just for the ratio above 1?

4. conclusions are too long, the are written as a overview of the resulñts; please integrate results and express the main conclusions

Author Response

Response to Reviewer 2 Comments

Point 1: Introduction did not indicate the state of the art in the field, and should be improved with more references and indicating the gaps in the knowledge and also the main novelty of the work; in this points, please differentiate with the previously reported article from a reference.

Response 1: Thank you very much for your valuable comments. After careful consideration, I now make the following reply.

Modified, see lines 86-89.

Point 2: In the material and method section, the3.2. Improved Groundwater Quality Index is not clear, please improve and also explain more its utility.

Response 2: Modified, see lines 177-190.

Point 3: Some results are not clear the way they were calculates, for example 230-233 "25% of the nitrate in the groundwater in the study area exceeded the standard",.... " the nitrate content of groundwater in this area exceeded the standard by 15% "in row 271-272 "...indicating that the nitrate content in the groundwater in the study area is carcinogenic and seriously threatens human health" I do not understand well if the formula is non-carcinogenic compounds how the authors express that the compound is carcinogenic, just for the ratio above 1?

Response 3: The limit value for exceeding the standard is for the Chinese national standard and the World Health Organization standard respectively.

According to American standards, more than 1 means that there is a non-carcinogenic risk.

Point 4: conclusions are too long, the are written as a overview of the resulñts; please integrate results and express the main conclusions

Response 4: Modified, see conclusion.

Reviewer 3 Report

This is basically a good and relevant paper.

Here are some specific comments that I hope you will find useful:

Some statements made need references (eg: The first sentence in the Abstract-I accept that some don’t like referencing in an Abstract…..not sure what this journal thinks).

Some sentences are too long and need to be shortened.

The English could be improved (eg lines 86-87).

Author Response

Thank you very much for your valuable comments. After careful consideration, I now make the following reply. The sentence of the article has been revised, see 86-87, etc. The journal has requirements for abstract citation and has not been modified.

Round 2

Reviewer 1 Report

In my opinion, the sources of nitrates have been analysed too briefly and no account has been taken of other factors that might have contributed to the increase in nitrate concentrations in water. One-off studies carried out in the short term cannot be the basis for drawing long-term conclusions. Hence my proposal to turn the nature of the work into a report.

the legend in figures 2 and 4 is too small. 

Author Response

Dear teacher,

Hello!

Thank you very much for your valuable comments. Regarding your suggestion, I consulted the publishing editor and thought it was necessary to resubmit the manuscript for review, if the paper was changed to a working report. In view of this, I still stick to the original idea without changing it.

The problem that the legends in Figure 2 and Figure 4 are too small has been revised, see the figure in the paper for details.

Thank you again for your valuable comments!

Salute to you!

Reviewer 2 Report

The authors take into account the reviewer suggestions

Author Response

Dear teacher,

Hello!

Thank you very much for your valuable comments.

Wish you a happy life!